# Design Rainfall Change of Rainwater Source Control Facility to Meet Future Scenarios in Beijing

**DOI:** 10.3390/ijerph20054355

**Published:** 2023-02-28

**Authors:** Xuerao Wang, Zhiyong Zhao, Zhiming Zhang, Meifang Ren, Thomas Sagris

**Affiliations:** 1Beijing Climate Change Response Research and Education Center, Beijing University of Civil Engineering and Architecture, Beijing 100044, China; 2Arup International Consultants (Shanghai) Company Limited, Shanghai 200031, China; 3China Academy of Urban Planning & Design, Beijing 100044, China

**Keywords:** design rainfall, rainwater source control, climate change, urbanization, sponge city

## Abstract

Rainwater source control facilities are essential to sponge city construction in China. Their size is determined based on historical rainfall data. However, with global warming and rapid urban development, rainfall characteristics have also changed, potentially leading to the failure of rainwater source- control facilities to manage surface water in the future. In this study, the design rainfall’s change and spatial distribution are analyzed using historical (1961–2014) observation rainfall data and future (2020–2100) projection data of three CMIP6 climate models. The results show that EC-Earth3 and GFDL-ESM4 project that future design rainfall will increase. EC-Earth3 projects a significant increase, while MPI-ESM1-2 projects that the design rainfall will decrease significantly. From the perspective of space, the design rainfall isoline in Beijing has always increased from northwest to southeast. In the historical period, the difference in design rainfall in different regions has reached 19 mm, and this regional heterogeneity shows an increasing trend in the future projection of EC-Earth3 and GFDL-ESM4. The difference in design rainfall in different regions is 26.2 mm and 21.7 mm, respectively. Therefore, it is necessary to consider future rainfall changes in the design of rainwater source control facilities. The relationship curve between the volume capture ratio (VCR) of annual rainfall and design rainfall based on the rainfall data of the project site or region should be analyzed to determine the design rainfall of the rainwater source control facilities.

## 1. Introduction

With global warming and rapid urban development, many problems caused by rainwater are becoming more and more prominent. Several concepts on how to manage rainwater have been put forth to address these problems. The concept of low-impact development (LID) was first proposed in Maryland in the United States [1]. Later, it was accepted and extended in Australia, the United Kingdom, and some European countries, forming the concept of water-sensitive urban design (WSUD) [2], sustainable urban drainage systems (SUDSs) [3], and green infrastructure (GI). China recently presented the “sponge city” concept for managing rainwater [4,5,6], which is based on idea of low impact development (LID) in the U.S. It implies that cities can be resilient like sponges, easily adjusting to environmental changes and coping with flooding caused by rain and water pollution [7,8]. Urban water problems can be solved by combining the facilities of source, midway, and end, significantly improving the efficiency of rainwater control and utilization. Rainwater source control is crucial for constructing sponge cities because it reduces total runoff, peak runoff, and runoff pollutants [9,10]. A logical configuration of rainwater source management facilities can help preserve as much of the original natural hydrological cycle in the development area as possible. Compared with the “annual rainfall control rate”, “water quality control volume”, and “rainfall field control rate” proposed by the United States in rainwater control management, China uses “the volume capture ratio (VCR) of the annual rainfall” as an essential indicator to evaluate rainwater control capacity. It obtains corresponding design rainfall using long-term historical rainfall records spanning at least 30 years to determine the size of the facility [11].

According to certain studies, local or worldwide rainfall structures have changed due to global climate change [12,13,14], varyingly breaking original rainfall laws and making the rainfall data unstable. Therefore, it is of great significance to analyze the characteristics and changes in rainfall [15,16]. In the 21st century, global rainfall and extreme rainfall are increasing [17,18,19]. Especially in the middle and high latitudes of the Northern Hemisphere, the intensity and frequency of rainfall have increased significantly. The Intergovernmental Panel on Climate Change (IPCC) projected that the impact of global climate change will persist until the end of the 21st century [20]. Some studies show that the mean and maximum values of rainstorm days, volume, and intensity in future scenarios in China are increasing [21], and the impact of extreme rainfall events will be more severe [22,23,24]. Lu et al. pointed out that extreme rainfall in China is increasing due to global climate change, and extreme rainfall events in the North China Plain are highly random [25]. Zhao et al. found that urban expansion increased the frequency of short-term rainfall events, and the urban rainfall center tended to move towards densely built areas [26]. According to research by Golroudbary et al., the cumulative total rainfall in Dutch urban areas increased more than in rural areas between 2011 and 2015 [27].

The proper sizing of rainwater source control facilities is based on historical rainfall records. Still, the effects of climate change may result in facilities that are no longer appropriate for future scenarios. Firstly, increased extreme rainfall events could make it more difficult for the rainwater system to function correctly. On the other hand, high design requirements for rainwater control facilities may be inefficient if rainfall declines. For the rainwater source control facility to remain highly reliable and cost-effective in the face of a changing environment, it is necessary to calculate the design rainfall in conjunction with the future rainfall projection. This article analyzes the variations in the VCR and design rainfall correspondence curves at each site under various climate models and future scenarios over different periods using Beijing as an example. A design rainfall contour map has been produced. With this guide, big cities should be able to decide on total control standards for sponge cities in various areas more logically and conveniently.

## 2. Data and Methodology

### 2.1. Overview of the Study Area

Located on the northwest edge of the North China Plain, Beijing is one of the largest cities in the world. The central urban area is flat, with mountains in the west, north, and northeast. The mountains are distributed in a circular pattern. The plain area is 6339 km^2^, and the mountain area is 10,072 km^2^. The topography is higher in the northwest and lower in the southeast. It is a semi-humid and semi-arid monsoon climate with a typical warm temperate. With an average annual rainfall of more than 600 mm, it is hot and rainy in the summer, and cold and dry in the winter. It is one of the regions with the most significant rainfall in North China, but the seasonal distribution is uneven. A total of 80% of the annual rainfall is concentrated in June, July, and August in summer.

Beijing, the capital of China, is a megacity with a population of more than 21 million. Due to topographical characteristics and varying degrees of urbanization, Beijing has the most rainfall in the east, followed by the south. While the north and west have relatively modest rainfall, the central metropolitan area is more prone to experiencing extreme rainstorms [28]. Beijing’s total annual rainfall has significantly decreased during the last few decades [29,30,31]. However, Beijing’s rainfall has increased during the city’s rapid urbanization [32]. Additionally, the average annual rainfall in Beijing’s central urban region is higher than in the nearby areas due to the effect of urbanization [30,33].

In Beijing’s urban areas, rapid urbanization has resulted in an annual increase in impervious surface area, wasting rainwater resources, and causing groundwater levels to drop [34]. Additionally, air and surface pollution levels have increased as a result of urbanization in Beijing. Rainwater interacts with airborne particles during the initial stages of rainfall, causing wet deposition. Rainwater picks up pollutants from the air when it falls to the ground and washes the city’s subsurface, continuously accumulating and carrying more pollutants with less runoff [35]. Most rainfall may be absorbed and used on-site with rainwater source control facilities. It may effectively solve Beijing’s rainwater issues by achieving peak cutting, delay, and purification goals, releasing the strain on the pipe network and end control facilities.

### 2.2. Sources of Data

This study adopts 24 h daily rainfall data from 8:00 to 8:00 of the next day at 20 stations (see Figure 1 for the distribution of the stations and see Table 1 for the location information of the stations) provided by the Beijing Meteorological Bureau (http://data.cma.cn/, accessed on 22 November 2021) from 1961 to 2014 as the observation data. The rainfall accuracy is 0.1 mm.

Global climate models (GCMs) are the most advanced and popular tools to project climate change and have been widely used [36]. Each model, however, has benefits and drawbacks when it comes to simulating various climate factors in various places. In other words, there is not a single generalized ideal climate model. GCMs have been evaluated by a large number of researchers in China. We know from existing studies that EC-Earth3, GFDL-ESM4, and MPI-ESM1-2 are the models considered by most studies to be the best performers for rainfall simulations in China [37,38,39,40,41]. In this paper, we selected EC-Earth3, GFDL-ESM4, and MPI-ESM1-2 climatic models of CMIP6 (https://esgf-node.llnl.gov/search/cmip6/, accessed on 22 November 2021) as historical daily simulated rainfall data (1961–2014) and future projection data (2020–2100) under low radiative forcing, medium radiative forcing, and high radiative forcing scenarios (SSP1-2.6, SSP2-4.5, and SSP5-8.5). The basic information of the three models is shown in Table 2.

### 2.3. Statistical Downscaling

Downscaling techniques can reduce the simulation bias of regional rainfall and enhance the simulation effect of the model by downscaling large-scale climatic information to regional-scale information. The widely used quantile-based matching method assumes stability and only uses the model’s cumulative distribution function (CDF) and observations in the baseline period. In contrast, this study uses an equidistant CDF matching method (EDCDFm) downscaling method based on CDF [42]. Based on the discrepancy between the observation and model CDFs for the training (baseline) period, this method incorporates and adjusts the model CDF for the projection period. The method precisely explains how a particular model’s distribution changes between the projection and baseline periods. The adjustment function is left unchanged for a given percentile because it is assumed that the difference between the model and the observation during the training period also applies in the future. However, the distinctions between historical and prospective CDFs are also considered. Therefore, the method can be written mathematically as follows:(1)$${\tilde{x}}_{m-p.adjst.}=x_{m-p}+F_{o-c}^{-1}\left(F_{m-p}\left(x_{m-p}\right)\right)-F_{m-c}^{-1}\left(F_{m-p}\left(x_{m-p}\right)\right)$$
where ${\tilde{x}}_{m-p.adjst.}$ is the adjusted rainfall data for future periods; $x_{m-p}$ is the pre-adjusted rainfall data for the future period; $F_{o-c}^{-1}$, $F_{m-c}^{-1}$ is an inverse function of the CDF of the observation (o) and model data (m); and $F_{m-p}$ is the CDF of the model data.

In order to validate the downscaled data, the historical annual rainfall distributions of the observed data and the three models’ downscaled data are shown in Figure 2. The distributions show a trend of increasing and then decreasing annual rainfall from northwest to southeast for both the historical and the downscaled data. The values of rainfall are also generally consistent.

The curves of the cumulative probability distribution of the observed data and the three models’ downscaled data are shown in Figure 3. The four curves basically coincide. Therefore, the downscaled model data can better reflect the actual situation and have certain applicability in Beijing, which can be used for subsequent calculations and simulations.

### 2.4. Volume Capture Ratio of Annual Rainfall and Design Rainfall

The volume capture ratio (VCR) adopted when constructing sponge cities in China is an important indicator to reflect rainwater source control facilities in terms of total rainfall control. It is the ratio of annual controlled rainfall volume per unit area to annual rainfall. It describes the inhibitory effect of source discharge reduction facilities on the outflow of rainwater runoff. The design rainfall is an important indicator to determine the size of rainwater source control facilities. Due to the spatial differences in rainfall characteristics, the design rainfall corresponds to different VCRs in different regions. According to the Technical Guide for Sponge City Construction, the calculation method of design rainfall is as follows: Firstly, the rainfall events with daily rainfall ≤ 2 mm shall be removed. Then, the daily rainfall values are sorted from small to large. Finally, the ratio of the rainfall less than the design rainfall in the total rainfall is calculated (for the part less than the design rainfall, the total rainfall is calculated according to the actual rainfall; for the part larger than the design rainfall, the total rainfall is calculated according to the design rainfall, and the cumulative sum is calculated). The rainfall (daily rainfall) corresponding to this ratio (VCR) is the design rainfall. The relationship between the VCR and the design rainfall is as follows:(2)$$\alpha =\frac{n(RF>DesRF)\times DesRF+\sum \;RF(2\;\mathrm{mm}<RF<DesRF)}{\sum \;RF(RF>2\;\mathrm{mm})\;}\times 100\%$$
where α is the VCR; *n* (*RF* > *DesRF*) is the number of the daily rainfall that exceeds the design rainfall; *DesRF* is the design rainfall corresponding to the VCR = *RF* (2 mm < *RF* < *DesRF*) is the daily rainfall greater than 2 mm and less than the design rainfall; and *RF* (*RF* > 2 mm) is the daily rainfall greater than 2 mm.

The spatial distribution of the design rainfall in different regions of Beijing for different VCRs can be obtained using the Kriging interpolation. Due to space limitation, this paper only discusses four situations, i.e., when the VCR is 70%, 75%, 80%, and 85%, and counts the variation in design rainfall in each area under these four conditions.

## 3. Results and Discussion

### 3.1. Analysis of Future Rainfall Characteristics in Beijing

Source control facilities significantly reduce runoff under medium and small rainfall conditions with the highest frequency, while the capacity to reduce the runoff under heavy rainfall conditions is weaker. Historical daily rainfall and projected daily rainfall from various models and scenarios at each station are classified. The result is shown in Figure 4. At each station, the total rainfall with daily rainfall of less than 2.0 mm accounts for a small proportion, which does not exceed 10% under historical and various models and scenarios. This rainfall produces minimal surface runoff. The total rainfall with daily rainfall between 2.0 mm to 33.6 mm accounts for a large proportion of the total rainfall at each station. Compared with historical rainfall, the proportion of rainfall decreases in EC-Earth3, is almost equal in GFDL-ESM4, and has a significant increase in MPI-ESM1-2. Although this rainfall can generate runoff, it can be generally controlled by rainwater source control facilities to ensure in situ absorption and utilization of rainwater. The rainwater source control facilities cannot wholly control the part of rainfall with daily rainfall greater than 33.6 mm. Some rainwater is discharged. This rainfall also accounts for a large proportion of the total rainfall. The projection of EC-Earth3 shows that this rainfall has increased to a certain extent, with an average increase of 7.58%, 8.05%, and 10.86% at 20 stations under the three scenarios, respectively. At the same time, MPI-ESM1-2 projects a significant reduction in rainfall relative to historical levels, with an average decrease of 19.29%, 16.52%, and 15.56% at 20 stations under the three scenarios, respectively. In EC-Earth3 and MPI-ESM1-2, the proportion of rainfall when daily rainfall is greater than 33.6 mm increases with the enhancement in radiative forcing. In GFDL-ESM4, the proportion of rainfall in medium forcing scenario SSP2-4.5 is the highest. Within this rainfall, about 33.6 mm of rainfall depth can be controlled by rainwater source control facilities, but the rest is difficult to control.

It can be seen from Figure 4 that the four meteorological gauged stations, Zhaitang, Tanghekou, Foyeding, and Yanqing, maintain differences in rainfall characteristics with other stations in model projection and historical observation rainfall. The total amount of rainfall exceeding the design rainfall is relatively small. The source facilities can easily control the rainfall instead of discharging it into the pipe network or terminal facilities. This part of the total rainfall of Mentougou and Shunyi stations accounts for a large proportion. Under the current design rainfall standard, the rainfall in these two regions is relatively difficult to control by the source facilities but is quickly discharged.

The spatial distribution of the annual total amount of each type of rainfall is shown in Figure 5. Due to the limited space and similar spatial distribution characteristics in each scenario, only historical rainfall and future rainfall under SSP2-4.5 scenarios are discussed here. It can be seen that the annual total amount of daily rainfall of less than 2 mm is higher in the northwest of Beijing. This rainfall will increase in future projections of the three models. The most significant increase is seen in the MPI-ESM1-2 model, with an increase of 26.0 mm at the Tanghekou station, the most significant increase of all the stations. In the GFDL-ESM4 model, the growth was the smallest, with the Changping station decreasing by 2.4 mm, and the other stations increasing by 0.8–11.9 mm. The annual total amount of daily rainfall between 2 mm and 33.6 mm occurs predominantly in the north of Beijing. This rainfall is projected to increase significantly in MPI-ESM1-2, with an increase of 66.4–143.9 mm at each station. It is almost unchanged in the projection of EC-Earth3. In the projection of GFDL-ESM4, the decrease is 22.5–66.4 mm, and it is only increased by 10.6 mm at the Haidian station. Rainfall with daily rainfall greater than 33.6 mm shows a trend of increasing from northwest to southeast. This rainfall increases significantly in the projection of EC-Earth3, with an increase of 80.2–157.5 mm. In the projection of MPI-ESM1-2, the reduction is noticeable, ranging from 48.0 mm to 129.9 mm.

### 3.2. Corresponding Relationship between the VCRs and the Design Rainfall and Its Future Uncertainty

The corresponding relationship between the VCRs and the design rainfall at 20 stations in Beijing is shown in Figure 6 and Table 3. Taking the VCR of 85% as an example, we can see significant differences in the design rainfall across Beijing. In the areas near stations such as Zhaitang, Tanghekou, Foyeding, and Yanqing, the design rainfall is only 21.4–23.5 mm. However, in Shunyi, Fangshan, Mentougou, Xiayunling, and some other stations, the design rainfall of rainwater source control facilities is required to be 38.0–40.4 mm. The current design rainfall of Beijing is 33.6 mm, and it is inappropriate to determine the overall design rainfall within Beijing according to this single value.

It can be seen from Figure 4 and Table 3 that the smaller the proportion of the rainfall with daily rainfall exceeding 33.6 mm, the smaller the design rainfall of the station, such as Zhaitang, Tanghekou, Foyeding, and Yanqing. The larger the proportion, the greater the design rainfall of the station, such as Shunyi, Fangshan, Mentougou, and Xiayunling. The corresponding relationship is also shown in the future rainfall projection in Table 4. Therefore, the design rainfall in Beijing is highly correlated with the proportion of heavy rainfall. 

The projected average design rainfall corresponding to the VCRs (70–85%) under different scenarios in the three models of each station in Beijing is shown in Table 4. By comparing Table 3 and Table 4, under the SSP1-2.6 scenario, the future design rainfall is about equivalent to the historical rainfall. Under the scenarios of SSP2-4.5 and SSP5-8.5, the design rainfall in the future is higher than in the past. The design rainfall at most stations increases with the enhancement of radiative forcing. Considering the average conditions of the three models and the three scenarios, the design rainfall should be reduced to 24.2–25.9 mm in Zhaitang, Tanghekou, Foyeding, and Yanqing, and be increased to 37.8–40.9 mm in Shunyi, Fangshan, and Mentougou.

Compared with the historical rainfall, the EC-Earth3 and GFDL-ESM4 models project that the future design rainfall will increase. EC-Earth3, in particular, projects a significant increase. Under the SSP5-8.5 scenario, the Xiayunling station will reach a peak growth of 16.4 mm. However, the projection of MPI-ESM1-2 shows that the design rainfall will decrease significantly, with the most significant drop of 17.2 mm at the Mentougou station under the SSP1-2.6 scenario. 

Considering the perspective of the cost and maintenance of an environment of a virtuous cycle of water in the watershed, the VCR should not be too large but a suitable value. The best rainwater control amount should be based on the standard that the rainwater discharge is close to the natural condition. In the Technical Guide for Sponge City Construction, the western and northern parts of Beijing are in Zone II, while the rest are in Zone III, and the corresponding VCRs are 80–85% and 75–85%, respectively. Beijing should refer to the above limit value of the VCR and determine the goal of total runoff control of each region according to local conditions. When there is a specific need for drainage and waterlogging prevention in the area, the maximum limit value can also be exceeded to better achieve the goal of total runoff reduction and waterlogging prevention. 

In this study, the VCR of 85%, which is recommended by the Ministry of Housing and Urban-Rural Development of the People’s Republic of China for Beijing, is selected as the condition for determining the design rainfall of rainwater source control facilities.

When the VCR is 85%, compared with the historical design rainfall, the projection of EC-Earth3 increases by 5.7–17.5 mm under three scenarios, the projection of MPI-ESM1-2 decreases by 5.8–18.1 mm, and the projection of GFDL-ESM4 fluctuates by −0.2 mm–8.9 mm. In general, there is uncertainty in future design rainfall, with a fluctuation range of −18.1 mm to +17.5 mm.

Figure 7 shows the box plot of the design rainfall (VCR = 85%) in the future. Changping has the largest quartile deviation of 23.5 mm. Zhaitanag has the lowest quartile deviation of 10.2 mm. No outliers occurred at any station. Among all the stations, the projection range of 13.9–32.7 mm appearing in Foyeding is the lowest, while the projection range of 22.6–56.3 mm appearing in Xiayunling is the greatest. The uncertainty is caused by the selection of GCMs, SSPs, and the downscaling approach. Considering the uncertainty and giving a range of values for the design rainfall will provide decision makers with more scientific guidance to help them develop different design plans. For example, lower values within the design rainfall range may be chosen for areas where cost is a key consideration, while higher values may be chosen for areas where control effect is a key consideration. If uncertainty is ignored and the design rainfall is set at a single value, this may result in wasted resources or an unsatisfactory control effect. Therefore, to make the rainwater source control facilities more effective, further and more in-depth research is required. From the perspective of space, the design rainfall in different regions of Beijing is quite different. Therefore, when determining the size of rainwater source control facilities, the relationship curve of “VCR–design rainfall” should be analyzed according to the rainfall data of the project location or region. This approach can be used as a guide to better achieve rainwater control goals and avoid resource waste.

### 3.3. Spatial Distribution of Design Rainfall of Rainwater Source Control Facilities

Due to space limitation, this study only shows the spatial distribution of the design rainfall corresponding to VCRs under the SSP2-4.5 scenario, as shown in Figure 8. It can be seen that the design rainfall in Beijing has always increased from northwest to southeast in the past and will continue to in the future. The peak of the design rainfall is in the vicinity of Fangshan District and Shunyi District, forming a belt distribution from southwest to northeast, similar to the annual average rainfall distribution. This characteristic is considered to be significantly related to natural factors, such as Beijing’s topography, because it formed before the city’s rapid urbanization and has remained unchanged throughout history and will do so in the future. As the Taihang Mountains and the Yanshan Mountains surround the northwest of Beijing, southeast wind prevails in summer. When mountains block the warm moisture flow, it is easy to form rainfall in the piedmont plain. However, comparing the historical and future projected design rainfall, taking the VCR = 85% as an example, the historical design rainfall in Beijing is 21.4–40.4 mm. The EC-Earth3 model projects design rainfall of 30.1–56.3 mm. The GFDL-ESM4 model projects it to be 25.8–47.5 mm. Furthermore, the MPI-ESM1-2 model projects it to be 14.4–25.7 mm. The projection results of the three models are quite different. This relates to the underlying data, the experimental design, the various parameters of the forcing, etc. The regional heterogeneity of rainfall in Beijing has an increasing trend in the EC-Earth3 and GFDL-ESM4 models. Many studies on the impact of urbanization on rainfall have shown that the urban heat island effect, urban building obstruction effect, and urban condensation nodule effect can influence rainfall characteristics [33,43,44]. Therefore, regional heterogeneity of rainfall in Beijing is likely to be caused by both natural factors and human activities.

## 4. Conclusions

The study used historical observation daily rainfall data of 20 meteorological gauged stations in Beijing from 1961 to 2014, historical simulated daily rainfall data (1950–2014) of the EC-Earth3, GFDL-ESM4, and MPI-ESM1-2 climate models, and the projected data under three scenarios (SSP1-2.6, SSP2-4.5, and SSP5-8.5). The spatial distribution differences in the relationship between the VCR and the design rainfall in different regions in the past and future are discussed. The following conclusions are drawn:
(1)The change in future rainfall will likely result in poor effectiveness and cost of rain source control facilities, whose size is calculated by the historical data. It is necessary to consider future rainfall when determining the design rainfall of facilities.(2)The proportion of heavy rainfall directly affects the efficiency of rainwater source control facilities. In the projection of EC-Earth3 and MPI-ESM1-2 models, the proportion of this part of rainfall increases with the enhancement of radiative forcing, which means that the control effect of source facilities may be weaker under the scenario of stronger radiative forcing.(3)As a megacity, Beijing has noticeable spatial differences in design rainfall corresponding to the same VCR between different regions. In the historical period, the design rainfall range in different Beijing regions is 19 mm. This regional heterogeneity shows an increasing trend in the future projections of EC-Earth3 and GFDL-ESM4.(4)The design rainfall in Beijing has always increased from northwest to southeast in the past and will continue to do so in the future. Therefore, when planning rainwater source control facilities, using the current 33.6 mm single standard of design rainfall will cause a waste of resources in parts of the northwest. At the same time, the southeast cannot achieve the required rainwater control goal. Considering the average conditions of the three models and the three scenarios, the design rainfall should be reduced to 24.2–25.9 mm in Zhaitang, Tanghekou, Foyeding, and Yanqing, and be increased to 37.8–40.9 mm in Shunyi, Fangshan, and Mentougou.

## Figures and Tables

**Figure 1 ijerph-20-04355-f001:**
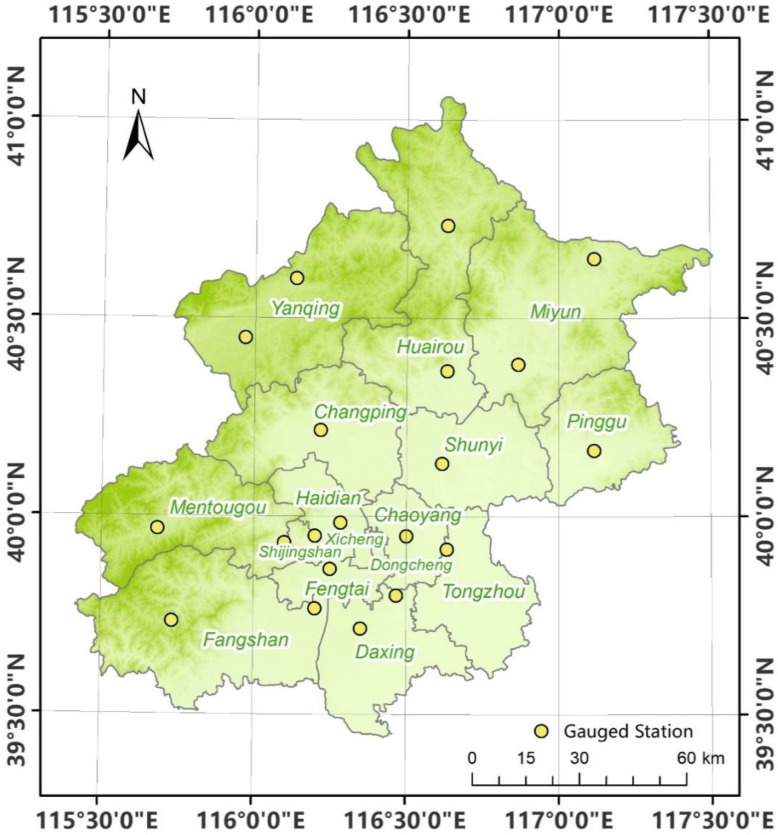
The location of 20 rain-gauged stations.

**Figure 2 ijerph-20-04355-f002:**
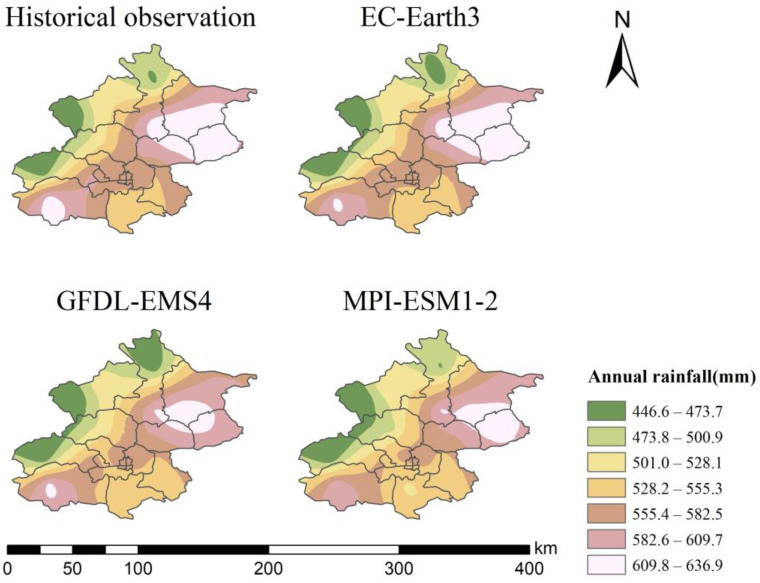
The historical annual rainfall distribution of the observed data and three models’ downscaled data.

**Figure 3 ijerph-20-04355-f003:**
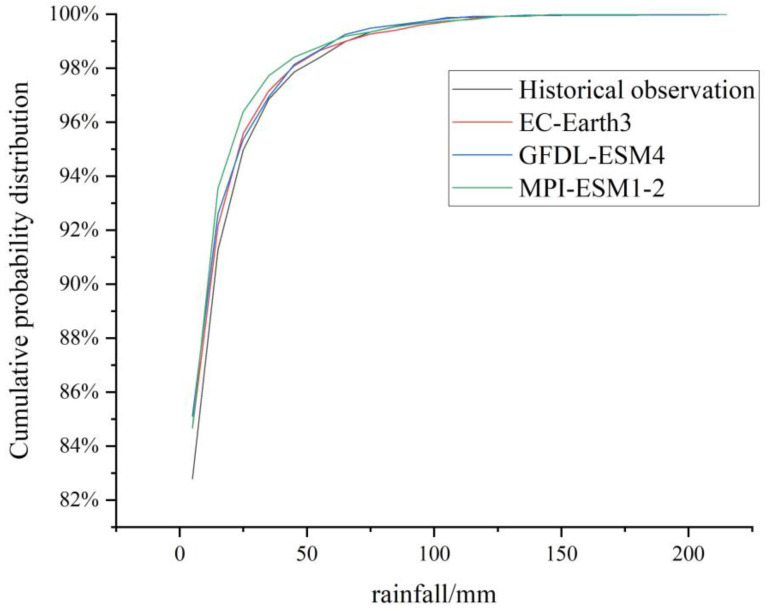
The curves of the cumulative probability distribution of the observed data and three models’ downscaled data.

**Figure 4 ijerph-20-04355-f004:**
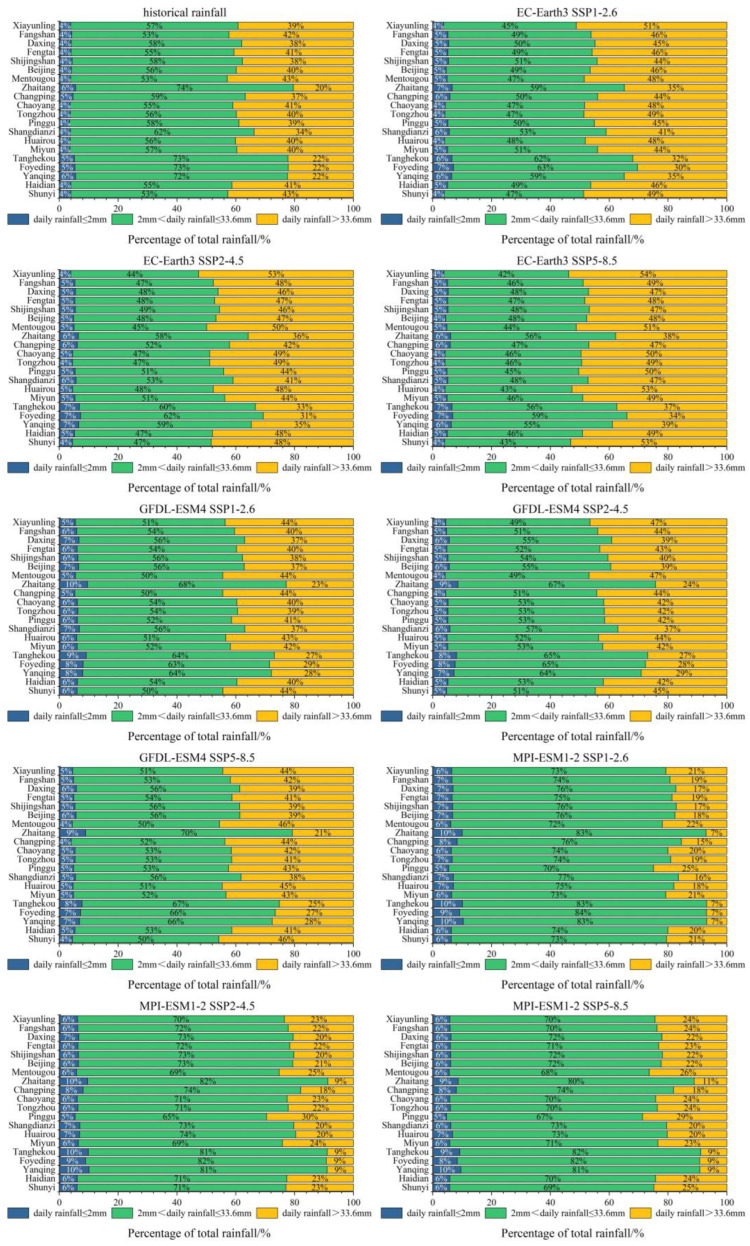
The proportion of different rainfall in Beijing.

**Figure 5 ijerph-20-04355-f005:**
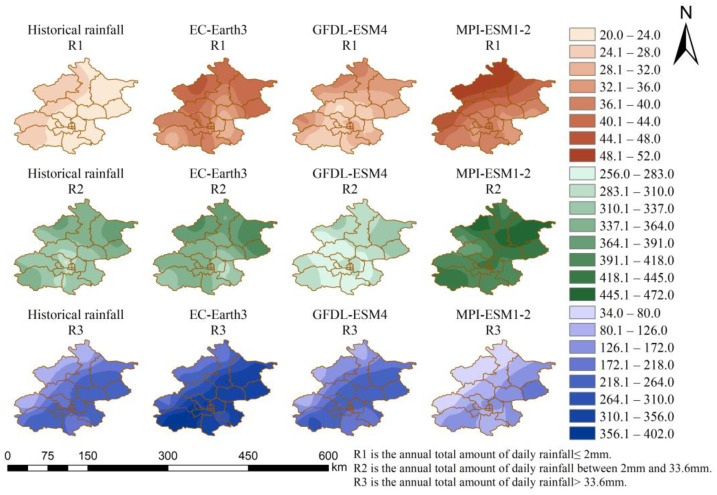
Spatial distribution of the annual total amount of each type of rainfall.

**Figure 6 ijerph-20-04355-f006:**
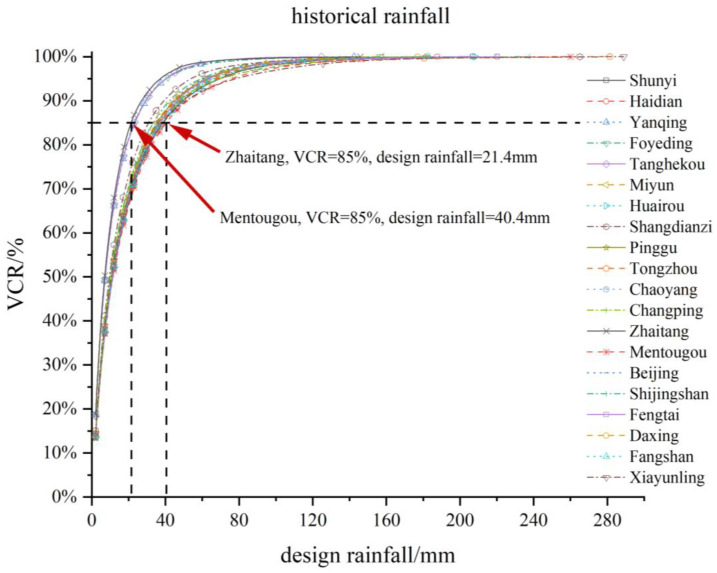
Correspondence between the VCR and the design rainfall in Beijing during historical periods.

**Figure 7 ijerph-20-04355-f007:**
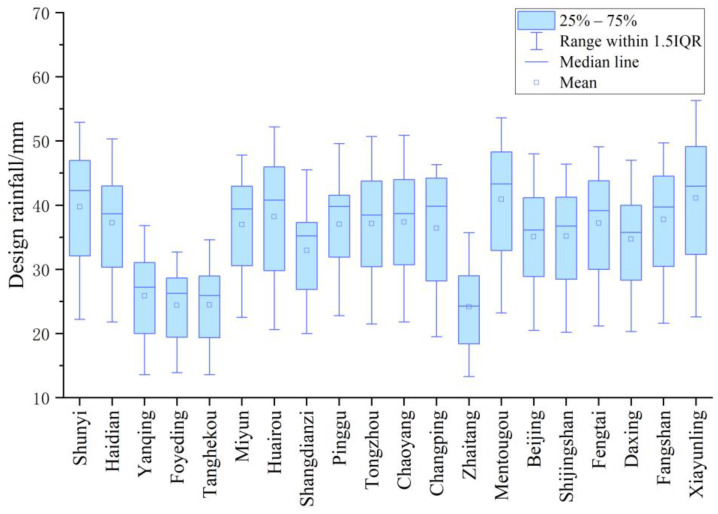
Box plot of the design rainfall (VCR = 85%) in the future.

**Figure 8 ijerph-20-04355-f008:**
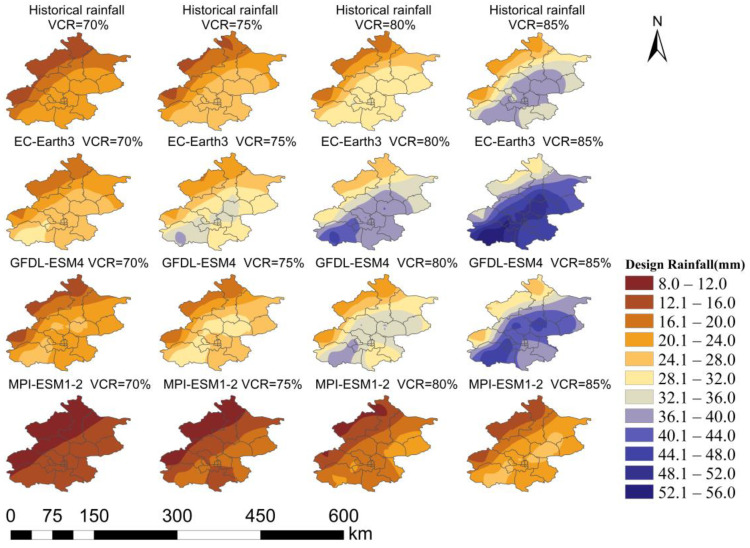
The spatial distribution of the design rainfall corresponding to the VCRs under the medium radiative forcing scenario SSP2-4.5.

**Table 1 ijerph-20-04355-t001:** The location information of 20 rain-gauged stations.

Station	Longitude (°E)	Latitude (°N)	Station	Longitude (°E)	Latitude (°N)
Shunyi	116.62	40.13	Chaoyang	116.50	39.95
Haidian	116.28	39.98	Changping	116.22	40.22
Yanqing	115.97	40.45	Zhaitang	115.68	39.97
Foyeding	116.13	40.60	Mentougou	116.12	39.92
Tanghekou	116.63	40.73	Beijing	116.47	39.80
Miyun	116.87	40.38	Shijingshan	116.20	39.95
Huairou	116.63	40.37	Fengtai	116.25	39.87
Shangdianzi	117.12	40.65	Daxing	116.35	39.72
Pinggu	117.12	40.17	Fangshan	116.13	39.68
Tongzhou	116.63	39.92	Xiayunling	115.73	39.73

**Table 2 ijerph-20-04355-t002:** Information of the CMIP6 models used in this study.

Model	Country and Area	Institution	Resolution (lon × lat)
EC-Earth3	Europe	EC-Earth Consortium	0.7° × 0.7°
GFDL-ESM4	USA	Geophysical Fluid Dynamics Laboratory	1.25° × 1°
MPI-ESM1-2	Germany	Max Planck Institute for Meteorology	0.94° × 0.94°

**Table 3 ijerph-20-04355-t003:** The historical design rainfall under different VCRs (70–85%) at each station in Beijing.

Station	70%	75%	80%	85%
Shunyi	22.3	26.4	31.6	38.7
Haidian	22	25.9	31.1	38.1
Yanqing	13.7	16.1	19.2	23.3
Foyeding	13.6	15.9	19	23.1
Tanghekou	13.6	16	19.1	23.5
Miyun	21.2	25.3	30.4	37.5
Huairou	21.8	25.9	31.2	38.7
Shangdianzi	18.3	21.5	25.6	30.9
Pinggu	20.9	24.6	29.3	35.6
Tongzhou	20.9	24.8	29.7	36.3
Chaoyang	21.8	25.7	30.6	37.3
Changping	19.9	23.8	29.1	36.5
Zhaitang	12.9	15.1	17.8	21.4
Mentougou	22.9	27.3	32.8	40.4
Beijing	20.9	24.6	29.4	35.8
Shijingshan	19.8	23.4	27.7	33.4
Fengtai	21.4	25.3	30.3	36.9
Daxing	20.5	24.2	28.9	35.1
Fangshan	21.9	26.1	31.2	38
Xiayunling	21.4	25.6	31.2	38.8

**Table 4 ijerph-20-04355-t004:** The projected average design rainfall corresponding to the VCRs (70–85%) under different scenarios in the three models of each station in Beijing.

	SSP1-2.6				SSP2-4.5				SSP5-8.5			
Station	70%	75%	80%	85%	70%	75%	80%	85%	70%	75%	80%	85%
Shunyi	21.7	26.0	31.4	38.7	22.0	26.5	32.1	39.8	23.0	27.4	33.1	40.7
Haidian	20.1	24.0	29.0	35.6	21.2	25.5	30.9	38.2	21.4	25.6	31.0	38.0
Yanqing	14.1	16.8	20.4	25.1	14.3	17.2	20.8	26.0	15.0	17.9	21.6	26.5
Foyeding	13.3	15.9	19.2	23.7	13.6	16.2	19.7	24.4	14.2	16.9	20.4	25.1
Tanghekou	13.3	15.9	19.3	23.8	13.6	16.3	19.7	24.3	14.5	17.2	20.7	25.3
Miyun	20.0	24.0	29.2	36.1	20.4	24.5	29.8	37.1	21.2	25.4	30.6	37.7
Huairou	20.8	25.0	33.7	37.5	21.0	25.2	30.6	38.1	21.9	26.2	31.6	39.0
Shangdianzi	17.8	21.3	25.8	31.5	18.2	21.9	26.4	32.9	19.5	23.2	28.0	34.4
Pinggu	20.4	24.2	29.3	35.9	20.9	25.0	30.4	36.0	22.2	26.4	31.9	39.3
Tongzhou	20.3	24.0	29.4	36.1	21.0	25.0	30.2	37.1	21.5	25.7	31.1	38.1
Chaoyang	20.4	24.4	29.6	36.3	21.1	25.3	30.4	37.4	21.5	25.8	31.2	38.4
Changping	19.8	23.8	28.8	35.3	20.1	24.2	29.6	36.9	20.7	24.7	29.9	36.9
Zhaitang	13.2	16.0	18.9	22.2	14.2	16.9	20.5	25.3	14.2	16.9	20.4	25.0
Mentougou	22.0	26.3	31.9	39.2	23.3	28.0	34.2	42.3	23.1	27.7	33.4	41.2
Beijing	19.1	22.9	27.7	34.2	19.8	23.7	28.5	35.0	20.3	24.3	29.3	36.0
Shijingshan	18.9	22.6	27.3	33.6	20.0	24.1	29.3	36.3	20.0	23.9	28.9	35.6
Fengtai	19.9	23.9	29.0	35.6	21.2	25.5	31.0	38.5	21.1	25.2	30.5	37.6
Daxing	18.7	22.3	27.1	33.1	19.8	23.7	28.8	35.5	20.0	23.9	28.9	35.5
Fangshan	20.3	24.3	29.4	36.2	21.6	25.9	31.5	39.0	21.4	25.6	31.0	38.1
Xiayunling	22.2	26.6	32.2	39.6	23.5	28.3	34.3	42.5	23.2	27.8	33.6	41.2

## Data Availability

Not applicable.

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
