# Peer review of "Design Rainfall Change of Rainwater Source Control Facility to Meet Future Scenarios in Beijing"

_ijerph, 2023, doi:10.3390/ijerph20054355_

Round 1

Reviewer 1 Report

This paper is well written and interesting. I only have two minor comments for authors to address:

1. The comparison between GCM simulated rainfall for the current climate conditions and historical observation is missing. This comparison is important because it can show the performance of GCM in capturing the current rainfall intensity over the study area, which improves the credibility of using such GCM for future climate.

2. The current “design rainfall” for the sponge city development is a single, deterministic, value, which does not account for uncertainty. However, uncertainty involves rainfall measurement, calculation of volume capture ratio, GCM, downscaling approach, and internal climate variability. Personally, I think the “design rainfall” should be a range of values instead of a single value. I would suggest that authors add a paragraph discussing the limitations of ignoring uncertainty in this “design rainfall” approach.

Reviewer 2 Report

This paper provides an in-depth examination of the change of "future" rainfall and its spatial distribution in Beijing, China in the near future, as predicted by three CMIP6 models. The authors have presented a well-crafted and comprehensible paper that is likely to be of great interest to engineers concerned with the design and optimization of Rainwater Source Control Facilities. Nevertheless, certain modifications should be made in order to refine the research and make it suitable for publication. My comments on this matter are detailed below.

It is clear that the forecast results for the future will carry a lot of uncertainty, even the results of CMIP6 products also carry a lot of uncertainty. I think the author could provide a forecast confidence interval of the future rainfall change rather than a single value. Moreover, for each CMIP6 product, the forecast results are also quite different. Could the author explain why?

- L124: The authors should elucidate the rationale for the selection of three distinct CMIP6 products in a more detailed manner.

- Table 2: What is the spatial resolution of the three products? What is the unit of “grid”?

- What is the accuracy of the rainfall data after downscaling? Did the author validate the downscaled data with the observed data?

- The ability to control runoff is highly dependent on rainfall intensity and duration, particularly in a short period of time. For example, 100m/day of rainfall will have a different impact than 100mm in a few hours. It is suggested that the author needs to analyze and discuss the frequency of future rainfall and the effect of heavy rainfall on Rainwater Source Control Facilities.

Round 2

Reviewer 1 Report

I appreciate the authors' response to my comments, which address all my previous questions. I will recommend this manuscript to be accepted in present form.

Reviewer 2 Report

All comments have been addressed. I think the paper can be considered for publication.